# The Preoperative Waiting Time on Long-Term Survival Following Elderly Hip Fracture Surgery

**DOI:** 10.3390/geriatrics10060153

**Published:** 2025-11-20

**Authors:** Chunyuan X. Qiu, Priscilla H. Chan, Kathryn E. Royse, Ronald A. Navarro, Glenn R. Diekmann, Kent T. Yamaguchi, Elizabeth W. Paxton, Vimal Desai

**Affiliations:** 1Department of Anesthesiology, Kaiser Permanente, Baldwin Park, CA 91706, USA; chunyuan.x.qiu@kp.org (C.X.Q.); vimal.desai@kp.org (V.D.); 2Medical Device Surveillance & Assessment, Kaiser Permanente, San Diego, CA 92108, USA; priscilla.chan@kp.org (P.H.C.); liz.w.paxton@kp.org (E.W.P.); 3Department of Orthopedics, Kaiser Permanente, Harbor City, CA 91706, USA; ronald.a.navarro@kp.org; 4Department of Orthopedics, Kaiser Permanente, Baldwin Park, CA 91706, USA; glenn.r.diekmann@kp.org; 5Department of Orthopedics, Kaiser Permanente, Santa Rosa, CA 95403, USA; kent.t.yamaguchi@kp.org

**Keywords:** elderly, hip fracture, trauma, mortality, anticoagulant, anesthesia

## Abstract

Background/Objective: The first-year postoperative mortality in elderly hip fracture patients is between 15 and 36%. Current scientific evidence indicates that morbidity and mortality are impacted by time of admission to surgery in hip fracture patients, although anticoagulation (AC) medication status specific optimization is unknown. Our objectives were to identify an ideal preoperative wait time by anticoagulation status in patients before hip fracture repair based on the incidence of postoperative morbidity and mortality. Methods: A total of 35,463 patients age ≥ 65 undergoing hip fracture repair were selected from a United States hip fracture registry (2009–2019). Patients were separated into strata (yes/no) based on whether they received anticoagulation (AC) medications ≤ 100 days prior to surgery. Multivariable logistic regression was adjusted for non-linear surgical wait time trends with prespecified percentiles using cubic splines. Results: A total of 87.1% (N = 30,902) of patients did not have AC preoperatively. Their median wait time was 20.3 h (IQR 13–27 h), and a positive linear trend was observed between surgical wait time and mortality. In patients with pre-operative AC, there was a “U”-shaped trend for all mortality time points although the breakpoint slopes were not significantly different from zero. Conclusions: In the study of more than 30,000 patients, short-term mortality was lowest for non-AC patients, undergoing surgery within the first 6–15 h of admission but remained uniform throughout the first 24 h of admission. These findings can be used to optimize patients prior to hip fracture surgery based on preoperative AC use and can positively affect resource planning and perioperative protocols.

## 1. Introduction

Hip fractures affect approximately 18% of elderly women and 6% of elderly men [1]. With the rapid growth of the aging population worldwide, the number of hip fractures is projected to reach 6.26 million by 2050 [2]. Despite advances in perioperative care and team-based management, both short- and long-term outcomes remain unsatisfactory, with first-year postoperative mortality ranging from 15 to 36%, and the total annual U.S. economic burden of hip fracture surgery exceeds $30 billion [3]. These trends highlight the urgent need for strategies that improve outcomes while managing costs in an aging population.

Preoperative waiting time is a modifiable factor associated with outcomes following hip fracture surgery [4,5]. Historically, a 72 h delay for medical stabilization was considered safe [4,5], but recent evidence favors early (<24 h), super early (<12 h), or ultra-early surgery (<6 h) [6]. Current guidelines, including AAOS, recommend surgery within 24–48 h of admission, citing potentially better clinical outcomes [7]. Systematic reviews and meta-analyses have linked early surgery to lower mortality and fewer perioperative complications [8]. However, these recommendations do not fully account for patient-specific considerations. The optimal timing may vary according to comorbidities, frailty, residual functional reserve, and tolerance to surgical and anesthesia trauma, while balancing the risks of ongoing deterioration and complications from immobilization [9,10].

Coagulopathies are a particularly complex factor in elderly hip fracture patients, often arising from cardiac arrhythmias or anticoagulation therapy [11]. Correcting coagulopathies must balance surgical bleeding risk, postoperative DVT/PE risk, and potential adverse effects on underlying cardiovascular disease [12,13]. Reversal strategies are time-consuming and lack standardized guidance [14]. Some patients undergo natural reversal over several days, while others receive active reversal to enable earlier surgery and potentially improved outcomes [15]. Evidence suggests continuing antiplatelets may be safe, and early surgery after discontinuation can reduce hospital stay without increasing adverse events [16]. Large trials, including HIP ATTACK, highlight the feasibility and safety of accelerated surgery, even though mortality benefits are not definitive [17]. Interventions to improve adherence to guideline-recommended timing and early mobilization show modest improvements in early outcomes, underscoring the potential value of system-level optimization [18,19].

Despite this growing evidence, there remains a lack of specific data on how preoperative anticoagulation modifies the relationship between surgical waiting time and mortality or complications. In this retrospective total joint registry study, we evaluated patients aged 65 and older who underwent hip fracture repair in an integrated healthcare system. Our objectives were to (1) identify whether preoperative anticoagulation influences short- (30-day, 90-day) and long-term (1-year) mortality in relation to waiting time for surgery and (2) determine how waiting time impacts the incidence of pneumonia, myocardial infarction, and venous thromboembolism in patients stratified by anticoagulation status.

## 2. Materials and Methods

### 2.1. Study Design and Data Source

We conducted a retrospective cohort study using a longitudinally maintained database from a US integrated healthcare system’s hip fracture registry. This registry includes all surgically treated hip fractures performed within the integrated healthcare system, which covers over 12.6 million members. Data collection, participation, and other details of hip fracture registry details have been previously published [20]. In brief, this registry identifies patients with surgically treated hip fractures and their demographics, medical comorbidities, intraoperative details, implant information, and outcomes using electronic medical records, administrative databases, and other institutional databases within the integrated healthcare system.

### 2.2. Inclusion and Exclusion Criteria

Patients aged 65 years and older who underwent surgery for a hip fracture between January 2009, and December 2019 were included. We did not include data on and after January 2020 due to COVID shelter-in-place policy that could influence surgical wait time and outcomes differently [21]. To maximize data integrity, we included only cases performed in the 4 main regions (Southern California, Northern California, Northwest, Hawaii) because they are mostly non-contract facilities.

Cases were excluded if they involved patients with pathological fractures, multiple fractures, open fractures (i.e., non-low energy falls or injuries from motor vehicle accidents), procedures in conjunction with other surgeries in the same hospital stay (i.e., complicated cases unrelated to the simple fall incidence), prior hip surgeries on the same side, surgery types other than nail fixation, pinning fixation, slidescrew fixation, or hemiarthroplasty. We also excluded cases if a patient was transferred from a different hospital to the hospital where the surgery took place (3% of patients), and those with missing surgical wait time (defined below, 0.8% of patients) (Figure 1).

### 2.3. Surgical Wait Time

Surgical wait time is defined as hours and minutes elapsed from either emergency department or inpatient admission to the surgical start time. The surgical start time is defined as a timestamp of the first document occurrence of when the patient was in the operating room, the procedure start time, or the incision time. All timestamps were documented in the electronic health record.

### 2.4. Outcomes of Interest

Primary outcomes were mortality at 30 days, 90 days, and 1 year. Secondary outcomes were 30-day pneumonia, 90-day myocardial infraction (MI), and 90-day venous thromboembolism (VTE).

Mortality information was obtained from the Social Security Administration, thus capturing all patients regardless of their insurance at the time of death. Medical complications included pneumonia, acute myocardial infarction (MI), deep venous thrombosis (DVT), and pulmonary embolism (PE), were defined according to the Agency for Healthcare Research and Quality (AHRQ) quality indicators [22]. DVT and PE were manually validated by clinical content experts to ensure database accuracy, including confirmation via ultrasound and/or computed tomography reports.

### 2.5. Anticoagulation Medication Before Surgery Leading to a Delay

Patients who have anticoagulation medications require medically necessary steps before surgery. For consistency with other pharmacoepidemiology studies and to reduce potential exposure misclassification bias, we defined anticoagulation users as those that received an anticoagulation medication dispensed in our integrated system within 100 days preop, including the following: aspirin, apixaban, betrixaban, edoxaban, fondaparinux, rivaroxaban, danaparoid, heparin flush, dalteparin, enoxaparin, tinzaparin, bivalirudin, dabigatran, desirudin, lepirudin, warfarin, abciximab, eptifibatide, tirofiban, cangrelor, caplacizumab, cilostazol, clopidogrel, dipyridamole, prasugrel, ticagrelor, ticlopidine, and vorapaxar [23,24]. We stratified patients by preop anticoagulation (AC) medication status (yes/no) in this study [3,4].

### 2.6. Covariates

Patient characteristics included age, sex (male vs. female), American Society of Anesthesiologists (ASA) score (III/IV/V vs. I/II), body mass index (BMI) (per 1 increment), smoking status (current, previous vs. never), diabetes (yes vs. no), and 90-day preop acute myocardial infraction (yes vs. no). We assessed the following Elixhauser’s comorbidities [5] (yes vs. no): chronic pulmonary disease, congestive heart failure, hypertension, psychoses, and renal failure. Surgical characteristics included anesthesia type (neuraxial, others, vs. general), surgery type (pinning, slidescrew, hemi vs. nail). We adjusted for variation observed in hospital volume [6] (low (0 to 124 cases per year), medium (125 to 186 cases per year), vs. high (≥187 cases per year), region (Southern California, Northwest, Hawaii vs. Northern California), and yearly trend (operative year per 1 year increment).

### 2.7. Statistical Analysis

Within each AC status strata, wait time was modeled flexibly using restricted cubic splines with knots at pre-specified percentiles of the distribution at 5, 27.5, 50, 72.5, and 95th percentiles [7,25] using restricted cubic spline and multivariable logistic regression models. Non-linearity was assessed with a likelihood ratio test. Model calibration and collinearity diagnostics confirmed adequate model fit and stability. We assessed any breakpoint for any change in slope with the Davies test [26] on data with wait times within 1st to 95th percentile. The longest 5% wait time was excluded at this step to minimize the influence of extreme wait time outliers on the linear trend from extrapolation. Fitted smooth curve vs. wait time, estimated breakpoint, and the slopes before and after breakpoint were presented.

Missing values were handled using mean imputation in the following covariates: (ASA *N* = 2114 (6.0%), BMI *N*= 201 (0.6%), smoking status *N*= 622 (1.8%), chronic pulmonary disease *N* = 8 (<0.1%), congestive heart failure *N* = 8 (<0.1%), hypertension *N* = 8 (<0.1%), psychoses *N* = 8 (<0.1%), renal failure *N* = 8 (<0.1%), and anesthesia type *N* = 160 (0.5%). Analyses were performed using R version 3.6.2. *p* < 0.05 was the statistical significance threshold used for this study, and all tests were 2-sided.

### 2.8. Sensitivity Analysis

We repeated the analysis and further stratified by the combination of AC groups and anesthesia type (general or neuraxial); we excluded ‘other’ as an anesthesia type group due to the difficulty of interpretation due to heterogeneity in this category.

## 3. Results

A total of 35,463 patients were identified in the study period, and procedures were performed by 546 surgeons at 35 hospitals. For the overall cohort, the median patient age was 83 years, and 30.2% were male (Table 1).

### 3.1. Patients Without Preoperative Anticoagulation Medications

87.1% (N = 30,902) of the total sample were not prescribed AC preoperatively; the median patient age was 83 years, 29.4% were male, and 74.0% had an ASA score of 3 or greater (Table 1). Median surgical wait time was 20.3 h (IQR: 13–27 h). The crude mortality rate was 5.1%, 10.7%, and 20.8% at 30 days, 90 days, and 1 year, respectively. A significant positive linear trend was observed after 15.5 h wait time for 30 day mortality (log odds ratio = 0.97, 95%CI 0.30–1.64), before 43.4 h wait for 90 days mortality (log odds ratio = 1.17, 95%CI 0.74–1.61), and after 6.1 h wait for 1 year mortality (log odds ratio = 0.87, 95%CI 0.57–1.16) (Table 2, Figure 2).

For 30-day pneumonia, an increased rate was observed before 43.9 h of wait (log odds ratio = 0.69, 95%CI 0.21–1.17). A flat line trend was observed for 90-day VTE and 90-day MI, with breakpoints at 15.3 and 14.3 h, respectively, though slopes before and after breakpoints were not significantly different from zero (Figure 3).

### 3.2. Patients with Preoperative Anticoagulation Medications

12.9% (*N* = 4561) of the total sample had AC preoperatively, with a median age of 84 years, 35.5% were male, and 89.7% had an ASA score of 3 or greater. Their median wait time was 26.2 h (IQR 18–44 h). The crude mortality rate was 8.3%, 16.3%, and 30.5% at 30 days, 90 days, and 1 year, respectively.

After controlling for covariates, the model fitted curve showed a “U”-shaped trend in 30 day, 90 day, and 1 year mortality rates vs. wait time, with the lowest at 22.5 h (Figure 1). Breakthrough point was observed at 20.5, 22.4, and 28.5 h of wait time for 30 days, 90 days, and 1 year mortality, respectively (Table 2). All slopes before the breakpoint were negatively correlated (i.e., higher mortality rate with shorter wait time before the breakpoint), while slopes after the breakpoint were positively correlated (i.e., higher mortality rate with longer wait time after the breakpoint), though slopes before and after the breakpoint were not significantly different from zero.

For 30 days of pneumonia vs. wait time, a “U”-shaped trend was observed with a significant increase after 16.9 wait hours (log odds ratio = 0.77 for every increment of 1 h wait, 95%CI 0.13–1.41). A flat line trend was observed for VTE and MI, with breakpoints at 68.7 and 16.2 h, respectively, though slopes before and after breakpoints were not significantly different from zero.

## 4. Discussion

In this retrospective cohort study using a well-established total joint registry from a large integrated healthcare system, we found that the timing of hip surgery can impact patient 90 day and 1 year mortality. In non-anticoagulated patients, the 90-day mortality progressively increased when the preoperative waiting time was more than 15.5 h. A shorter preoperative waiting time (approximately 6.1 h) was associated with better 1-year survival. In anticoagulated patients, early or late surgery was associated with higher mortality, with the inflection point at 22.4 h for 90-day mortality and 28.5 h for 1-year mortality. While earlier surgical intervention can benefit some patients, especially those without coagulopathy, this approach may not be desirable for anticoagulated patients. We found that optimal preoperative waiting time was not a fixed point but a dynamic variable.

Many guidelines support early surgical intervention, defined as within 24–48 h. However, a shorter preoperative waiting time has been suggested or even preferred, as it may reduce length of hospital stay, morbidity, and mortality [18]. In a retrospective Canadian study involving 42,230 patients, surgery within 24 h demonstrated significantly lower 30-day mortality, and surgery within 6 h may benefit patients even more. However, this observation was not substantiated by the HIP ATTACK study, which was limited to outcomes up to 90 days following surgery [19]. Using long-term survival data in this study, we found that ultra-early surgical intervention close to 6 h in non-anticoagulated hip fracture patients was associated with reduced 90 day and 1 year mortality, but only in non-anticoagulated patients. This finding of long-term survival benefits for early surgery is notable because causes of death in the perioperative period and later stages often differ. Unlike 30 day mortality, which is often due to pneumonia, sepsis, and MI, later deaths are often due to decompensating chronic diseases such as cardiovascular disease, cancer, aging, or neurocognitive deterioration. Increasing evidence suggests that patients who experience eventful and prolonged perioperative courses, especially in the presence of delirium, are at accelerated risk of physical, functional, and cognitive decompensation and death in the later stage. In non-anticoagulated patients, early surgical interventions, often characterized by more frequent use of regional anesthesia and analgesia, can facilitate early ambulation, feeding, and rehabilitation. Additionally, regional anesthesia and analgesia can provide more effective pain control with reduced narcotic use. We previously reported that more frequent use of regional anesthesia and analgesia is associated with lower perioperative mortality.

The presence of anticoagulants in elderly hip fracture patients is considered a warning sign for higher perioperative morbidities and mortalities, occurring in approximately 10% of patients. This warning often reflects advanced cardiovascular disease such as atrial fibrillation, valvular disease, CHF, or recent MI. Co-management of cardiovascular disease and coagulopathy in these patients is extremely challenging. First, current guidelines for perioperative management of anticoagulants for elective surgery are often not applicable to urgent hip fracture patients; second, the process of anticoagulation to coagulation reversal to anticoagulation carries risk because both bleeding and thrombotic events can occur simultaneously. Consequently, practices are heterogeneous with varied outcomes. In our study, patients with coagulopathy had higher mortality than patients without anticoagulation (16.3% vs. 10.7% for 90-day mortality, 30.5% vs. 20.8% for 1-year mortality). Patients on anticoagulants also waited longer for surgery compared to the non-anticoagulant group (35.7 ± 33.3 vs. 23.8 ± 20.0 h), which is shorter for both groups compared to literature (47 vs. 29 h) [16]. Interestingly, preoperative waiting time in anticoagulated hip fracture patients exhibited a biphasic pattern for 90-day and 1-year mortality. The “U-shaped” curve, where early or late surgery was associated with higher mortality, implies that stress accommodation or resistance to secondary trauma imposed by mandatory surgery and anesthesia can change quickly, and the window of opportunity is narrowed in this vulnerable group. A similar pattern was observed for postoperative MI and pneumonia, with an understandable absence of VTE.

This study derives from a well-recognized, longitudinally maintained total joint registry based on a comprehensive electronic medical record rather than an administrative database from an integrated healthcare delivery system. To our knowledge, this is the first study to illustrate the dynamic characteristics of preoperative waiting time on long-term mortality and morbidity following elderly hip fracture surgery.

Our study has several limitations. Although we adjusted for many potential confounders, residual confounding due to unmeasured variables may remain due to the retrospective design. We used admission time rather than fractured time, which is often unknown. The cause of death was not investigated, which may further elucidate the impact of preoperative waiting time. Preoperative waiting time may also serve as a surrogate for the quality of patient-centered multidisciplinary teamwork rather than simple surgical decisions. Additionally, this study lacks external validation, and institutional factors—including operating room (OR) availability, surgical scheduling, and anesthesia provider availability—may have influenced outcomes.

## 5. Conclusions

Preoperative waiting time following an elderly hip fracture is linked to long-term survival. However, this window of opportunity for optimal outcomes is short and dynamic, dependent on baseline medical conditions and anticoagulation status. In non-anticoagulated patients, the window is between 6 and 15 h. In anticoagulated patients, it is shorter, between 22 and 29 h. These findings, which should be confirmed by future research studies, highlight the importance of individualized timing for hip fracture surgery, considering medical comorbidities, anticoagulation status, and institutional factors such as OR and anesthesia provider availability, to optimize both short- and long-term outcomes.

## Figures and Tables

**Figure 1 geriatrics-10-00153-f001:**
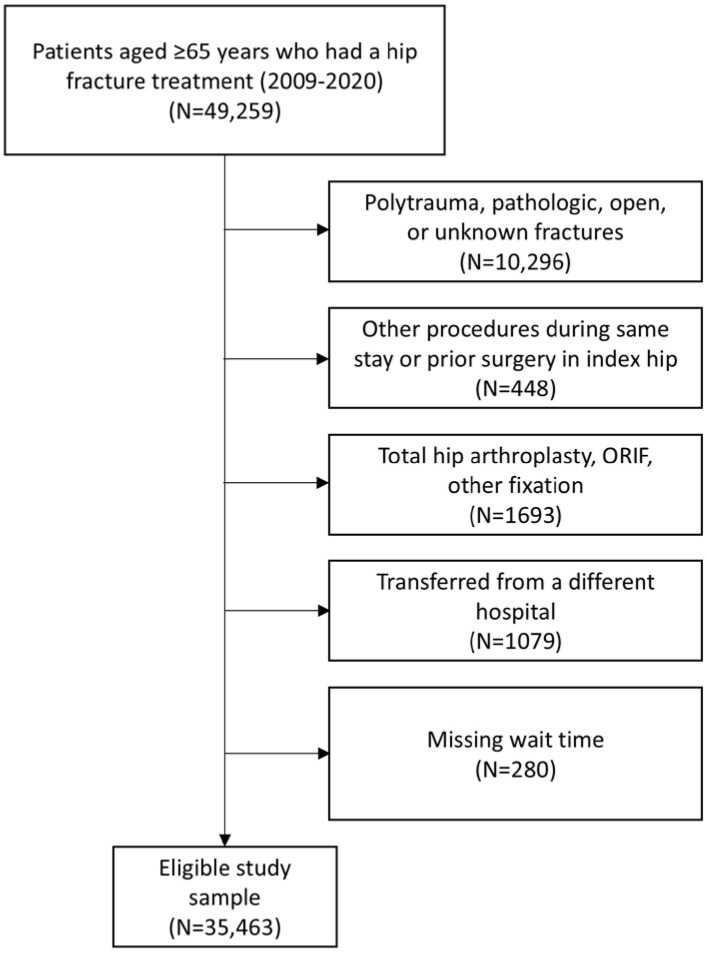
Patient inclusion and exclusion flowchart.

**Figure 2 geriatrics-10-00153-f002:**
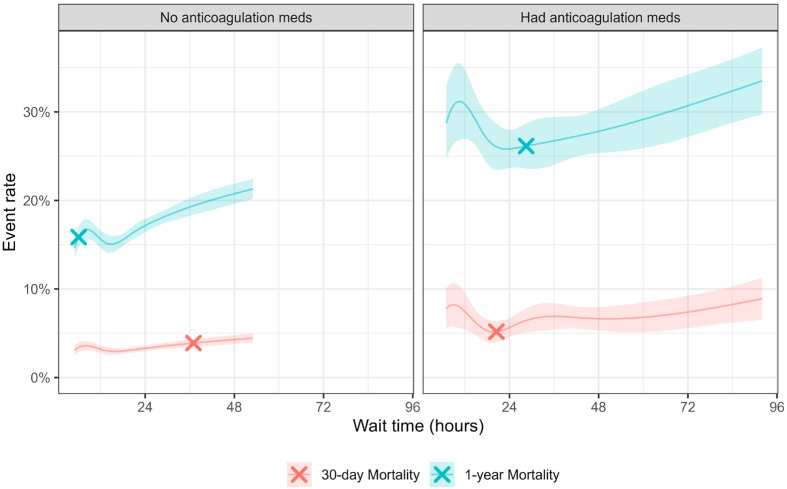
Mortality and wait time adjusted for covariates. Showing a smoothed curve and breakpoint within 5th percentile to the 95th percentile, wait hours to avoid extrapolation. Shades denoted the 95% confidence band. “X” denoted breakpoint.

**Figure 3 geriatrics-10-00153-f003:**
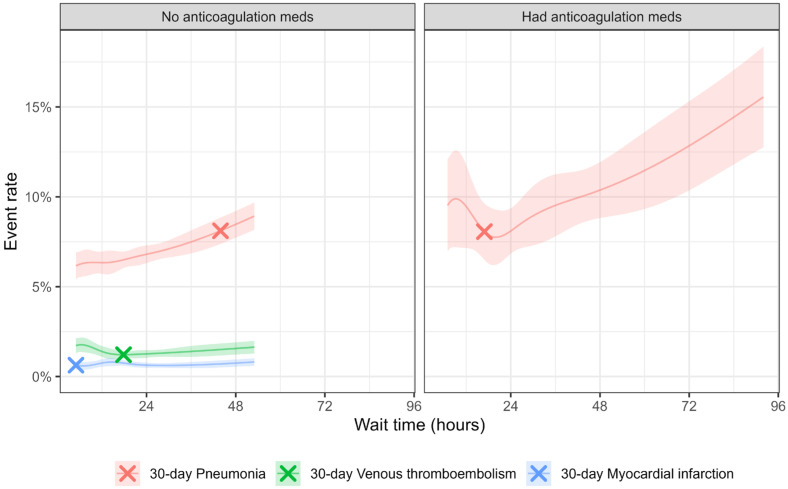
Outcomes and wait time adjusted for covariates. Showing smoothed curve and breakpoint within 5th percentile to 95th percentile wait hours to avoid extrapolation. Shades denoted the 95% confidence band. “X” denoted breakpoint.

**Table 1 geriatrics-10-00153-t001:** Characteristics of 35,463 patients who underwent hip fracture repair in a US-integrated healthcare system (January 2009–February 2020).

Characteristics, n (%) Unless Specified	No Anticoagulation Medication N = 30,902	Had Anticoagulation Medication N = 4561
Patient factors		
Age, years. Median (IQR) Male	83 (77–89) 9092 (29.4%)	84 (78–88) 1620 (35.5%)
ASA: III/IV/V	21,551 (74.0%)	3794 (89.7%)
BMI, kg/m^2^ Median (IQR) Smoking status	23.3 (20.6–26.5)	23.7 (21.0–27.0)
Never	16,264 (53.6%)	2215 (49.0%)
Quit smokers	12,110 (39.9%)	2060 (45.6%)
Current smokers	1946 (6.4%)	246 (5.4%)
90-day preop acute MI	191 (0.6%)	251 (5.5%)
Diabetes	8583 (27.8%)	1599 (35.1%)
Chronic pulmonary disease	7718 (25.0%)	1458 (32.0%)
Congestive heart failure	5032 (16.3%)	1640 (36.0%)
Hypertension	23,941 (77.5%)	3939 (86.4%)
Psychoses	4280 (13.9%)	827 (18.1%)
Renal failure	9965 (32.3%)	1921 (42.1%)
Surgical factors		
Anesthesia type		
General	16,734 (54.4%)	2895 (63.9%)
Neuraxial	13,214 (42.9%)	1509 (33.3%)
Converted to general	473 (1.5%)	59 (1.3%)
Monitored anesthesia care (MAC)	216 (0.7%)	48 (1.1%)
Regional only	132 (0.4%)	23 (0.5%)
Procedure		
Nail fixation	12,679 (41.0%)	1776 (38.9%)
Pinning fixation	3825 (12.4%)	622 (13.6%)
Slidescrew fixation	2392 (7.7%)	347 (7.6%)
Hemi-arthroplasty	12,006 (38.9%)	1816 (39.8%)
Hospital volume		
Low ≤ 125 cases/year	8365 (27.1%)	1300 (28.5%)
Medium 126–187 cases/year	8892 (28.8%)	1309 (28.7%)
High > 187 cases/year	13,645 (44.2%)	1952 (42.8%)
Surgical wait time, hours Mean ± SD Percentile: 5%, 25%, 50%, 75%, 95%	23.8 ± 20.0 5.1, 13.2, 20.3, 27.4, 53.3	35.7 ± 33.3 6.9, 17.9, 26.2, 44.3, 91.7

IQR = interquartile range; ASA = American Society of Anesthesiologists; BMI = body mass index; MI = myocardial infarction; SD = standard deviation.

**Table 2 geriatrics-10-00153-t002:** Outcomes and wait time breakpoints in 35,463 patients who underwent hip fracture repair in a US integrated healthcare system (January 2009–February 2020).

Outcome	Frequency (%)	Breakpoint Wait Time in Hours	Before Breakpoint Slope (95%CI)	After Breakpoint Slope (95%CI)
Patients with no pre-op anticoagulation medications (N = 30,902)				
30 days mortality	1500 (5.1%)	15.5	−0.58 (−3.38, 2.22)	**0.97 (0.30, 1.64)**
90 days mortality	3309 (10.7%)	43.4	**1.17 (0.74, 1.61)**	−2.64 (−8.03, 2.75)
1 year mortality	6428 (20.8%)	6.1	8.01 (−3.67, 19.69)	**0.87 (0.57, 1.16)**
30 days pneumonia	2519 (8.2%)	43.9	**0.69 (0.21, 1.17)**	−0.47 (−7.03, 6.09)
90 days VTE	771 (2.5%)	15.3	−1.84 (−5.44, 1.76)	0.44 (−0.55, 1.43)
90 days MI	506 (1.6%)	14.3	2.59 (−3.35, 8.53)	−0.77 (−2.00, 0.45)
Patients with pre-op anticoagulation medications (N = 4561)				
30 days mortality	378 (8.3%)	20.5	−1.90 (−5.79, 1.99)	0.50 (−0.30, 1.31)
90 days mortality	742 (16.3%)	22.4	−1.18 (−3.60, 1.24)	0.50 (−0.16, 1.15)
1 year mortality	1390 (30.5%)	28.5	−0.55 (−1.90, 0.79)	0.47 (−0.23, 1.18)
30 days pneumonia	449 (10.9%)	16.9	−1.42 (−6.51, 3.67)	**0.77 (0.13, 1.41)**
90 days VTE	87 (1.9%)	68.7	−0.36 (−1.86, 1.14)	7.42 (−3.31, 18.15)
90 days MI	169 (3.7%)	16.2	−4.70 (−14.64, 5.24)	0.62 (−0.53, 1.78)

Note: Slope was in logit (% outcome) with 1 increment of wait hour. VTE = venous thromboembolism. MI = myocardial infraction. Bolded values indicate statistical significance *p* < 0.05.

## Data Availability

The data presented in this study are available on request from the corresponding author. Data used for this study is unavailable due to privacy or ethical restrictions.

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
