# Peer review of "The Preoperative Waiting Time on Long-Term Survival Following Elderly Hip Fracture Surgery"

_geriatrics, 2025, doi:10.3390/geriatrics10060153_

Round 1
Reviewer 1 Report
Comments and Suggestions for Authors
Points for improvement
1. Title and abstract
The abstract is overly dense and written in a somewhat cumbersome style.
2. Introduction
The literature review is extensive but somewhat repetitive in places and includes outdated references (for example, cost estimates from 2002).
The gap in knowledge should be clarified: the lack of specific evidence on how anticoagulation modifies the relationship between surgical waiting time and mortality.
It is advisable to introduce an explicit hypothesis and a closing paragraph that clearly defines the primary and secondary objectives.
3. Methods
The retrospective design is well described, but a more detailed description of the registry (data quality criteria, rate of missing data) would be useful.
The definition of “anticoagulation within the previous 100 days” is debatable; the authors should justify this time frame or perform a sensitivity analysis using an alternative threshold (e.g., 30 days).
It is not clearly stated how intraoperative or peri-hospital deaths were handled.
Mean imputation for missing values is suboptimal; multiple imputation methods or sensitivity analyses are recommended.
Details are lacking regarding the verification of statistical assumptions (multicollinearity, model fit).
Confidence intervals and exact p-values should be included in the main figures or tables.
4. Results
This section is clear, but the tables are lengthy and contain some repetitive variables; a flow diagram summarising case exclusions would improve readability.
Figures 1 and 2 require better legibility (axes and units should be more visible).
It is recommended to summarise the breakpoints numerically in an additional plot (e.g., forest plot).
5. Discussion
Direct comparisons with equivalent studies (e.g., HIP ATTACK or recent meta-analyses on early surgery) are missing.
The authors should explore in greater depth the physiological mechanisms explaining differences between anticoagulated and non-anticoagulated patients (bleeding effects, haemodynamic stability, pharmacological reversal).
Interpretation of the “U-shaped” curve should be supported by visual confidence intervals or supplementary tables that substantiate the observed trend.
Although the discussion of limitations is appropriate, it would be useful to mention the lack of external validation and the potential influence of institutional factors (surgical scheduling, anaesthetist availability).
6. Conclusions
Clear and coherent, but they should focus more on clinical applicability—namely, practical recommendations regarding safe surgical timing windows according to anticoagulant use.
The need for prospective studies to confirm the optimal time frames identified should be emphasised.
Author Response
Comments and Suggestions for Authors
Points for improvement
- Title and abstract
The abstract is overly dense and written in a somewhat cumbersome style.
Author Response: Thank you for your comments. We have modified the abstract to fit within the current guidelines of 250 words.
- Introduction
The literature review is extensive but somewhat repetitive in places and includes outdated references (for example, cost estimates from 2002).
It is advisable to introduce an explicit hypothesis and a closing paragraph that clearly defines the primary and secondary objectives.
Author response: We have removed the 2002 reference following information from the introduction and have included instead an annual projection of hip fracture related costs for the US:
(Lines 40-41) …”and the total annual U.S. economic burden of hip fracture surgery exceeds $30 billion [3].”
Author response: We have also included the primary and secondary objectives
(Lines 72-75) “Our objectives were to (1) identify whether preoperative anticoagulation influences short-(30-day, 90-day) and long-term (1-year) mortality in relation to waiting time for surgery, and (2) determine how waiting time impacts the incidence of pneumonia, myocardial infarction, and venous thromboembolism in patients stratified by anticoagulation status.”
The gap in knowledge should be clarified: the lack of specific evidence on how anticoagulation modifies the relationship between surgical waiting time and mortality.
Author response: Thank you for your comments. We have added information to the paragraph in the introduction on the relevance of assessing time to separately by anticoagulation status.
(Lines 55-67) “Coagulopathies are a particularly complex factor in elderly hip fracture patients, often arising from cardiac arrhythmias or anticoagulation therapy [11]. Correcting coagulopathies must balance surgical bleeding risk, postoperative DVT/PE risk, and potential adverse effects on underlying cardiovascular disease [12,13]. Reversal strategies are time-consuming and lack standardized guidance [14]. Some patients undergo natural reversal over several days, while others receive active reversal to enable earlier surgery and potentially improved outcomes [15]. Evidence suggests continuing antiplatelets may be safe, and early surgery after discontinuation can reduce hospital stay without increasing adverse events [16]. Large trials, including HIP ATTACK, highlight the feasibility and safety of accelerated surgery, even though mortality benefits are not definitive [17] Interventions to improve adherence to guideline-recommended timing and early mobilization show modest improvements in early outcomes, underscoring the potential value of system-level optimization [18,19].”
- Methods
The retrospective design is well described, but a more detailed description of the registry (data quality criteria, rate of missing data) would be useful.
Author response: Thank you for your comments. We have briefly covered information about the integrated hip fracture registry and unfortunately due to space constraints we are unable to provide more detailed information on the registry but instead have provided prior publications where this information can be accessed.
(Lines 80-87) “This registry includes all surgically treated hip fractures performed within the integrated healthcare system, which covers over 12.6 million members. Data collection, participation, and other details of hip fracture registry details have been previously published [1]. In brief, this registry identifies patients with surgically treated hip fractures and their demographics, medical comorbidities, intraoperative details, implant information, and outcomes using electronic medical records, administrative databases, and other institutional databases within the integrated healthcare system.”
We have also provided detailed information on inclusions and exclusion by indication in the flowchart and the methodology.
(Lines 95-102):”Cases were excluded if they involved patients with pathological fracture, multiple fractures, open fractures (i.e. non-low energy fall, or injured from motor vehicle accident), procedures in conjugation with other surgeries in the same hospital stay (i.e. complicated cases unrelated to the simple fall incidence), prior hip surgeries on the same side, surgery type other than nail fixation, pinning fixation, slidescrew fixation, or hemi arthroplasty. We also excluded cases if a patient was transferred from a different hospital to the hospital where the surgery took place (3% of patients), and those with missing surgical wait time (defined below, 0.8% of patients). (Figure 1)”
Figure 1. Patient inclusion and exclusion flowchart (could not be cut and pasted in this window but is updated in the document).
The definition of “anticoagulation within the previous 100 days” is debatable; the authors should justify this time frame or perform a sensitivity analysis using an alternative threshold (e.g., 30 days).
Author response: Our selection of 100 day window was based on both prior pharmacoepidemiology literature, anticoagulation effects that may carry over longer in drugs like warfarin, and prescribing patterns. Most oral anticoagulation medications are dispensed in 30-, 60- and 90-day supplies so including a brief gap prior to surgery ensures that individuals who had the medication within 100 days of surgery are properly identified as prior users. A shorter window could potentially misclassify long-term users who refill less frequently and could potentially introduce exposure misclassification bias.
Pharmacoepidemiologic studies often use 90-120 days to define current use of chronic medications as this prior balances the half-life, refill, and adherence patterns typically included in claims data. We have now included this information in the methods section as well as two references.
(Lines 124-131) “For consistency with other pharmacoepidemiology studies and to reduce potential exposure misclassification bias we defined anticoagulation users as those that received an anticoagulation medication dispensed in our integrated system within 100 days preop, including: aspirin, apixaban, betrixaban, edoxaban, fondaparinux, rivaroxaban, danaparoid, heparin flush, dalteparin, enoxaparin, tinzaparin, bivalirudin, dabigatran, desirudin, lepirudin, warfarin, abciximab, eptifibatide, tirofiban, cangrelor, caplacizumab, cilostazol, clopidogrel, dipyridamole, prasugrel, ticagrelor, ticlopidine, vorapaxar.”
References:
Abdel-Qadir H, Austin PC, Pang A, Fang J, Udell JA, Geerts WH, McNaughton CD, Jackevicius CA, Kwong JC, Yeh CH, Cox JL, Lee DS, Ko DT, Atzema CL. The association between anticoagulation and adverse outcomes after a positive SARS-CoV-2 test among older outpatients: A population-based cohort study. Thromb Res. 2022 Mar;211:114-122. doi: 10.1016/j.thromres.2021.12.010. Epub 2021 Dec 13. PMID: 35149396; PMCID: PMC8667561.
Tanskanen A, Taipale H, Koponen M, Tolppanen AM, Hartikainen S, Ahonen R, Tiihonen J. Drug exposure in register-based research-An expert-opinion based evaluation of methods. PLoS One. 2017 Sep 8;12(9):e0184070. doi: 10.1371/journal.pone.0184070. PMID: 28886089; PMCID: PMC5590868.
It is not clearly stated how intraoperative or peri-hospital deaths were handled.
Author response: We have added this information to the methods section.
Mean imputation for missing values is suboptimal; multiple imputation methods or sensitivity analyses are recommended.
Author response: Because the proportion of missingness for adjustment covariates was <5% except for ASA and patterns were consistent with Missing completely at random/missing at random given observed data we used mean imputation to preserve sample size and minimize complexity. Prior literature has shown that when missingness is minimal (<5%) a complete case or simple imputation yields a negligible bias. As MI primarily requires correct specification and improves model efficiency, which we were not concerned with due to our large sample size.
References:
White IR, Carlin JB. Bias and efficiency of multiple imputation compared with complete-case analysis for missing covariate values. Stat Med. 2010 Dec 10;29(28):2920-31. doi: 10.1002/sim.3944. PMID: 20842622.
Sterne JA, White IR, Carlin JB, Spratt M, Royston P, Kenward MG, Wood AM, Carpenter JR. Multiple imputation for missing data in epidemiological and clinical research: potential and pitfalls. BMJ. 2009 Jun 29;338:b2393. doi: 10.1136/bmj.b2393. PMID: 19564179; PMCID: PMC2714692.
Details are lacking regarding the verification of statistical assumptions (multicollinearity, model fit).
Author response: Thank you for your comment. Due to the clinical nature of the journal some of the statistical information was not included in the manuscript although we did conduct model diagnostics for functional form checks via the likelihood ratio test (LRT), prespecified the know percentiles a priori based on standard recommendations, centered the reference value which was median wait time to aid in interpretation of the intercept and reduce collinearity, and assessed multicollinearity by inspecting the variance inflation factors for the spline basis plus included covariates. For simplification we have added this information to the statistical analysis section.
(Lines 145-149): “Within each AC status strata, wait time was modeled flexibly using restricted cubic splines with knots at pre-specified percentiles of the distribution at 5, 27.5, 50, 72.5, 95th percentile [7] using restricted cubic spline and multivariable logistic regression models. Non-linearity was assessed with a likelihood ratio test. Model calibration and collinearity diagnostics confirmed adequate model fit and stability.”
Confidence intervals and exact p-values should be included in the main figures or tables.
Author response: Thank you for your comment. We believe there may be some confusion about figures 2 and 3 included in the manuscript. The goal of spline figures is to illustrate the functional form of the association between wait time and mortality/morbidity while adjusting for covariates rather than to present statistical tests. Because the model is a single continuous logistic regression with restricted cubic splines the overall uncertainty is already reflected in the shaded 95% confidence bands around those curves. Adding separate p-values or confidence intervals at each point would be misleading as the estimates across the spline are not independent and multiple comparisons issues would arise.
In Table 2, as listed in the footnotes of the table, the slope represents the change in the log-odds of the outcome (equivalent to the percentage change in event probability on the logit scale) per 1-hour increase in wait time. Values are derived from multivariable logistic regression models which have been adjusted for covariates with separate slopes estimated before and after the identified breakpoint. 95% confidence intervals are show here as well to indicate the precision of estimates and p-values are not presented because the intervals fully convey statistical uncertainty.
- Results
This section is clear, but the tables are lengthy and contain some repetitive variables; a flow diagram summarising case exclusions would improve readability.
Figures 1 and 2 require better legibility (axes and units should be more visible).
It is recommended to summarise the breakpoints numerically in an additional plot (e.g., forest plot).
Author response: We have already included a figure flow diagram in the manuscript which is copied below. We understand the point about legibility and have increased the dpi of all figures included in the text which are copied below for your review. We don’t understand the suggestion to summarize the breakpoints in an additional plot as we have included all information for the breakpoints in Figures 2 and 3 and Table 2 where we have transformed the log-odds to the logit scale around the breakpoints for ease of comparison. We have standardized and updated the formatting of tables 1 and 2 but feel that it is important to include all covariates assessed.
Figure 1. Patient inclusion and exclusion flowchart.
Figure 2. Mortality and wait time adjusted for covariates. Showing smoothed curve and breakpoint within 5th percentile to 95th percentile wait hours to avoid extrapolation. Shades denoted the 95% confidence band. “X” denoted breakpoint.
Figure 3. Outcomes and wait time adjusted for covariates. Showing smoothed curve and breakpoint within 5th percentile to 95th percentile wait hours to avoid extrapolation. Shades denoted the 95% confidence band. “X” denoted breakpoint.
- Discussion
Direct comparisons with equivalent studies (e.g., HIP ATTACK or recent meta-analyses on early surgery) are missing.
Author comment: We have provided discussion of equivalent studies including HIP ATTACK but have expanded upon these in both the introduction and discussion.
(Lines 229-235) “Many guidelines support early surgical intervention, defined as within 24–48 hours. However, a shorter preoperative waiting time has been suggested or even preferred, as it may reduce length of hospital stay, morbidity, and mortality [18]. In a retrospective Canadian study involving 42,230 patients, surgery within 24 hours demonstrated significantly lower 30-day mortality, and surgery within 6 hours may benefit patients even more. However, this observation was not substantiated by the HIP ATTACK study which was limited to outcomes up to 90 days following surgery [19]. Using long-term survival data in this study, we found that ultra-early surgical intervention close to 6 hours in non-anticoagulated hip fracture patients was associated with reduced 90-day and 1-year mortality, but only in non-anticoagulated patients. This finding of long-term survival benefits for early surgery is notable because causes of death in the perioperative period and later stages often differ.”
The authors should explore in greater depth the physiological mechanisms explaining differences between anticoagulated and non-anticoagulated patients (bleeding effects, haemodynamic stability, pharmacological reversal).
Author comment We do provide a discussion of some of the differences between patient groups but believe that a discussion beyond what is listed below is beyond the scope of this paper and would distract from the relevance and discussion of our findings.
(Lines 245-255) “In non-anticoagulated patients, early surgical interventions, often characterized by more frequent use of regional anesthesia and analgesia, can facilitate early ambulation, feeding, and rehabilitation. Additionally, regional anesthesia and analgesia can provide more effective pain control with reduced narcotic use. We previously reported that more frequent use of regional anesthesia and analgesia is associated with lower perioperative mortality. The presence of anticoagulants in elderly hip fracture patients is considered a warning sign for higher perioperative morbidities and mortalities, occurring in approximately 10% of patients. This warning often reflects advanced cardiovascular disease such as atrial fibrillation, valvular disease, CHF, or recent MI. Co-management of cardiovascular disease and coagulopathy in these patients is extremely challenging.”
Interpretation of the “U-shaped” curve should be supported by visual confidence intervals or supplementary tables that substantiate the observed trend.
Author response: We have addressed the comments regarding visual confidence intervals already as figures 2 and 3 both have 95% confidence bands and Table 2 lists 95% confidence intervals.
Although the discussion of limitations is appropriate, it would be useful to mention the lack of external validation and the potential influence of institutional factors (surgical scheduling, anaesthetist availability).
Author response: We have detailed the limitations mentioned above in discussion.
(Lines 276-284) “Our study has several limitations. Although we adjusted for many potential con-founders, residual confounding due to unmeasured variables may remain due to the retrospective design. We used admission time rather than fractured time, which is often unknown. The cause of death was not investigated, which may further elucidate the impact of preoperative waiting time. Preoperative waiting time may also serve as a surrogate for the quality of patient-centered multidisciplinary teamwork rather than simple surgical decisions. Additionally, this study lacks external validation, and institutional factors—including operating room (OR) availability, surgical scheduling, and anesthesia provider availability—may have influenced outcomes.
- Conclusions
Clear and coherent, but they should focus more on clinical applicability, namely, practical recommendations regarding safe surgical timing windows according to anticoagulant use.
The need for prospective studies to confirm the optimal time frames identified should be emphasised.
Author response: Thank you for your suggestion. We have incorporated the need for future research to confirm our findings but believe the distinction between prospective and restrospective is arbitrary given the outcomes are likely not to be affected by recall bias. We have already listed the clinical windows as an actionable window for surgery but feel further discussion is outside of the scope of our investigation
(Lines 286-294) “However, this window of opportunity for optimal outcomes is short and dynamic, dependent on baseline medical conditions and anticoagulation status. In non-anticoagulated patients, the window is between 6 and 15 hours. In anticoagulated patients, it is shorter, between 22 and 29 hours. These findings, which should be confirmed by future research studies, highlight the importance of individualized timing for hip fracture surgery, considering medical comorbidities, anticoagulation status, and institutional factors such as OR and anesthesia provider availability, to optimize both short- and long-term outcomes.”

Reviewer 2 Report
Comments and Suggestions for Authors
Dear Authors,
Below you may find my comments regarding your manuscript:
The Abstract is very long and does not follow the indications given by the journal; it has 646 words instead of 250, therefore it must be reviewed. The Keywords should include elderly or another similar word, as your study refers to this age group.
In the Introduction the authors explained the importance of their research and gave enough information based on significant published data. They refer to the impact of the waiting time before surgery for hip fracture on the clinical outcome, discussing also the role of comorbidities, such as coagulopathies and the need for preoperative correction, in delaying surgery. Please clarify the first question of your aim of the study (lines 110-112) as in my opinion the explanation is long, confusing and difficult to understand.
Please replace Methods with Materials and Methods, according to the Manuscript preparation criteria. In Study Design and Data Source you refer to an article published in 2015 regarding „Data collection, participation, and other details of hip fracture registry details..” In my opinion you can mention this in Discussions but I do not see the relevance of this information here. Please present more details about inclusion criteria and clarify what type of surgical interventions were evaluated, as one can only speculate it from the exclusion criteria paragraph. No results should be given here (see lines 140-141). Please relocate Fig.1 which presents to the selection process, as it is misplaced and added after a paragraph that has no connection with its information. The rest of the section is presenting sufficient data regarding the evaluation of the selected study group and the statistical analysis is complex and appropriate.
In Results section, the information from the study group was included in two categories: patients with and without preoperative anticoagulant treatment. For each group all the evaluated parameters were presented in a clear and systematic manner.
In Discussion the authors begin with the presentation of their main findings which are further compared to representative available data. In my opinion you should answer more clearly to the questions formulated in the final paragraph of the Introduction section; the discussion is focused in the influence of the time interval before surgery while the information regarding the second question is too limited. For such a retrospective study based on a large cohort and a 10 years interval, the conclusions might deserve a special section. This could increase the clinical impact of your data.
The References must be presented in the order of appearance in the text, not alphabetically, as this is clear requirement from the journal.
Author Response
Below you may find my comments regarding your manuscript:
Author response: Thank you for your thorough review and comments. Our color coded responses are below in blue with updates in red.
The Abstract is very long and does not follow the indications given by the journal; it has 646 words instead of 250, therefore it must be reviewed. The Keywords should include elderly or another similar word, as your study refers to this age group.
Author Response: Thank you for your comments. We have modified the abstract to fit within the current guidelines of 250 words. We have also included the keyword elderly which is a great suggestion.
(Lines 32-33) Keywords: elderly, hip fracture; trauma; mortality; anticoagulant; anesthesia
In the Introduction the authors explained the importance of their research and gave enough information based on significant published data. They refer to the impact of the waiting time before surgery for hip fracture on the clinical outcome, discussing also the role of comorbidities, such as coagulopathies and the need for preoperative correction, in delaying surgery. Please clarify the first question of your aim of the study (lines 110-112) as in my opinion the explanation is long, confusing and difficult to understand.
Author Response: Thank you we have simplified our objective statements and both are listed below.
(Lines 72-75) “Our objectives were to (1) identify whether preoperative anticoagulation influences short-(30-day, 90-day) and long-term (1-year) mortality in relation to waiting time for surgery, and (2) determine how waiting time impacts the incidence of pneumonia, myocardial infarction, and venous thromboembolism in patients stratified by anticoagulation status.”
Please replace Methods with Materials and Methods, according to the Manuscript preparation criteria.
Author Response:Updated
(Line 77) 2. Materials and Methods
In Study Design and Data Source you refer to an article published in 2015 regarding „Data collection, participation, and other details of hip fracture registry details..” In my opinion you can mention this in Discussions but I do not see the relevance of this information here.
Author Response: This information is listed in the materials and methods as it is relevant to describing the hip fracture registry in more detail rather than a discussion of the results as related to other research
Please present more details about inclusion criteria and clarify what type of surgical interventions were evaluated, as one can only speculate it from the exclusion criteria paragraph.
Author Response: We have included a detailed section on inclusion and exclusion criteria and flowchart for review. The only included cases were for nail fixation
(Lines 95-102)” Cases were excluded if they involved patients with pathological fracture, multiple fractures, open fractures (i.e. non-low energy fall, or injured from motor vehicle acci-dent), procedures in conjugation with other surgeries in the same hospital stay (i.e. complicated cases unrelated to the simple fall incidence), prior hip surgeries on the same side, surgery type other than nail fixation, pinning fixation, slidescrews fixation, or hemi arthroplasty. We also excluded cases if a patient was transferred from a different hospital to the hospital where the surgery took place (3% of patients), and those with missing surgical wait time (defined below, 0.8% of patients). (Figure 1)
Figure 1. Patient inclusion and exclusion flowchart.
No results should be given here (see lines 140-141). Please relocate Fig.1 which presents to the selection process, as it is misplaced and added after a paragraph that has no connection with its information. The rest of the section is presenting sufficient data regarding the evaluation of the selected study group and the statistical analysis is complex and appropriate.
Author response: We have reformatted the manuscript and moved the figures and tables to when they were first mentioned in the text. No results are presented in the materials and methods section only information related to the missingness of covariates.
In Results section, the information from the study group was included in two categories: patients with and without preoperative anticoagulant treatment. For each group all the evaluated parameters were presented in a clear and systematic manner.
Author response: Thank you for your comments.
In Discussion the authors begin with the presentation of their main findings which are further compared to representative available data. In my opinion you should answer more clearly to the questions formulated in the final paragraph of the Introduction section; the discussion is focused in the influence of the time interval before surgery while the information regarding the second question is too limited. For such a retrospective study based on a large cohort and a 10 years interval, the conclusions might deserve a special section. This could increase the clinical impact of your data.
Author response: Thank you we have updated the discussion and conclusions based on your recommendations.
Discussion
(Lines 217-294) In this retrospective cohort study using a well-established total joint registry from a large integrated healthcare system, we found that the timing of hip surgery can impact patient 90-day and 1-year mortality. In non-anticoagulated patients, the 90-day mortal-ity progressively increased when the preoperative waiting time was more than 15.5 hours. A shorter preoperative waiting time (approximately 6.1 hours) was associated with better 1-year survival. In anticoagulated patients, early or late surgery was asso-ciated with higher mortality, with the inflection point at 22.4 hours for 90-day mortality and 28.5 hours for 1-year mortality. While earlier surgical intervention can benefit some patients, especially those without coagulopathy, this approach may not be desirable for anticoagulated patients. We found that optimal preoperative waiting time was not a fixed point but a dynamic variable.
Many guidelines support early surgical intervention, defined as within 24–48 hours. However, a shorter preoperative waiting time has been suggested or even preferred, as it may reduce length of hospital stay, morbidity, and mortality [18]. In a retrospective Canadian study involving 42,230 patients, surgery within 24 hours demonstrated sig-nificantly lower 30-day mortality, and surgery within 6 hours may benefit patients even more. However, this observation was not substantiated by the HIP ATTACK study which was limited to outcomes up to 90 days following surgery [19]. Using long-term survival data in this study, we found that ultra-early surgical intervention close to 6 hours in non-anticoagulated hip fracture patients was associated with reduced 90-day and 1-year mortality, but only in non-anticoagulated patients. This finding of long-term survival benefits for early surgery is notable because causes of death in the perioperative period and later stages often differ. Unlike 30-day mortality, which is often due to pneumonia, sepsis, and MI, later deaths are often due to decompensating chronic dis-eases such as cardiovascular disease, cancer, aging, or neurocognitive deterioration. Increasing evidence suggests that patients who experience eventful and prolonged perioperative courses, especially in the presence of delirium, are at accelerated risk of physical, functional, and cognitive decompensation and death in the later stage. In non-anticoagulated patients, early surgical interventions, often characterized by more frequent use of regional anesthesia and analgesia, can facilitate early ambulation, feeding, and rehabilitation. Additionally, regional anesthesia and analgesia can provide more effective pain control with reduced narcotic use. We previously reported that more frequent use of regional anesthesia and analgesia is associated with lower perioperative mortality.
The presence of anticoagulants in elderly hip fracture patients is considered a warning sign for higher perioperative morbidities and mortalities, occurring in approximately 10% of patients. This warning often reflects advanced cardiovascular disease such as atrial fibrillation, valvular disease, CHF, or recent MI. Co-management of cardiovascular disease and coagulopathy in these patients is extremely challenging. First, current guidelines for perioperative management of anticoagulants for elective surgery are of-ten not applicable to urgent hip fracture patients; second, the process of anticoagulation to coagulation reversal to anticoagulation carries risk because both bleeding and thrombotic events can occur simultaneously. Consequently, practices are heterogeneous with varied outcomes. In our study, patients with coagulopathy had higher mortality than patients without anticoagulation (16.3% vs. 10.7% for 90-day mortality, 30.5% vs. 20.8% for 1-year mortality). Patients on anticoagulants also waited longer for surgery compared to the non-anticoagulant group (35.7 ± 33.3 vs. 23.8 ± 20.0 hours), which is shorter for both groups compared to literature (47 vs. 29 hours) [16]. Interestingly, preoperative waiting time in anticoagulated hip fracture patients exhibited a biphasic pattern for 90-day and 1-year mortality. The “U-shaped” curve, where early or late surgery was associated with higher mortality, implies that stress accommodation or resistance to secondary trauma imposed by mandatory surgery and anesthesia can change quickly, and the window of opportunity is narrowed in this vulnerable group. A similar pattern was observed for postoperative MI and pneumonia, with an under-standable absence of VTE.
This study derives from a well-recognized, longitudinally maintained total joint registry based on a comprehensive electronic medical record rather than an administrative da-tabase from an integrated healthcare delivery system. To our knowledge, this is the first study to illustrate the dynamic characteristics of preoperative waiting time on long-term mortality and morbidity following elderly hip fracture surgery.
Our study has several limitations. Although we adjusted for many potential con-founders, residual confounding due to unmeasured variables may remain due to the retrospective design. We used admission time rather than fractured time, which is often unknown. The cause of death was not investigated, which may further elucidate the impact of preoperative waiting time. Preoperative waiting time may also serve as a surrogate for the quality of patient-centered multidisciplinary teamwork rather than simple surgical decisions. Additionally, this study lacks external validation, and insti-tutional factors—including operating room (OR) availability, surgical scheduling, and anesthesia provider availability—may have influenced outcomes.
- Conclusions
Preoperative waiting time following an elderly hip fracture is linked to long-term sur-vival. However, this window of opportunity for optimal outcomes is short and dynamic, dependent on baseline medical conditions and anticoagulation status. In non-anticoagulated patients, the window is between 6 and 15 hours. In anticoagulated patients, it is shorter, between 22 and 29 hours. These findings, which should be confirmed by future research studies, highlight the importance of individualized timing for hip fracture surgery, considering medical comorbidities, anticoagulation status, and institutional factors such as OR and anesthesia provider availability, to optimize both short- and long-term outcomes.
The References must be presented in the order of appearance in the text, not alphabetically, as this is clear requirement from the journal.
Author response: We apologize for the issues with formatting and have updated the references to be standard with the request for order of appearance and MDPI journals.